# Ecological Adaptability of Some Cultivars and Breeding Samples of *Origanum vulgare* L.

**Elena Myagkikh** [1,*]**, Svetlana Babanina** [1] **, Alexander Mishnev** [1]**, Ludmila Radchenko** [1]**, Vladimir Pashtetskiy** [1]**, Natalya Nevkrytaya** [1] **and Olga Loretts** [2]

1 Research Institute of Agriculture of Crimea, Kievskaya St., 150, 295453 Simferopol, Russia; svetlana.babanina@bk.ru (S.B.); avmishnev@mail.ru (A.M.); l-radchenko@ukr.net (L.R.); pvs98a@gmail.com (V.P.); nevkritaya@mail.ru (N.N.)

2 Faculty of Biotechnology and Food Engineering, Ural State Agrarian University, Karla Liebknechta St., 42, 620075 Yekaterinburg, Russia; rector.urgau@yandex.ru

\* Correspondence: origanum.science@mail.ru; Tel.: +8-978-7213839

**Abstract:** Since the registry of common oregano (*Origanum vulgare* L.) cultivars does not involve regionalization, a comprehensive study of cultivars bred by different institutions in the intended cultivation region is valuable and relevant. The objective of the research was to assess the possibility of using various indices of ecological adaptability originally developed for grain crops for their use in the most adapted genotypes' selection (breeding samples and cultivars) of *Origanum vulgare* L. to the temperate climate of the Crimean Peninsula. The research was carried out in the piedmont zone of Crimea from 2016 to 2019. The study material consisted of breeding samples No. 10 and No. 82 from the collection of FSBSI "Research Institute of Agriculture of Crimea", as well as cultivars Zima, Raduga, and Slavnitsa selected by the All-Russian Scientific Research Institute of Medicinal and Aromatic Plants (ARSRIMAP). Genotype had the greatest influence on yield of fresh oregano material (43%) with the influence of the weather conditions of the year being 2%. On the contrary, meteorological conditions had a much greater effect on the essential oil accumulation and its areal yield, which were 30% and 25%, respectively. In terms of the coefficient of ecological variation of fresh yield, sample No. 82 and Slavnitsa cultivar were the best (11.47–16.7%). The local genotypes No. 10 and No. 82 varied less by the essential oil content and its yield. The genotype effect value was greater than 0 in the Raduga cultivar and local genotype No. 82 for the yield, but only in No. 82 genotype for the other two characteristics. Cultivars Zima and Raduga were classified as intensive ($b_i > 1$) by the environmental flexibility of fresh yield, while local genotype No. 82 and Slavnitsa cultivar formed the group of intensive ones by essential oil content and essential oil yield. Local genotypes No. 10 and No. 82 were better than the introduced cultivars in terms of essential oil content homeostability and essential oil yield (*Hom* = 1.91–2.18). Thus, local genotypes proved to be more adapted to the region's conditions in terms of essential oil accumulation. However, they were inferior to the registered cultivars of ARSRIMAP breeding in terms of fresh yield.

**Keywords:** oregano (*Origanum vulgare* L.); cultivar; yield; essential oil content; oil yield; environmental flexibility; adaptability; homeostability; selective value

## 1. Introduction

Oregano (*Origanum vulgare* L.) is a valuable food and herb crop. Flowering above-ground parts and leaves of oregano are traditionally widely used in cooking and food industry as a flavoring agent for various dishes and beverages, and as an antioxidant in cosmetics. Currently, *O. vulgare* plants with high carvacrol content are also frequently used in poultry and livestock farming as nutritional supplements [1]. In addition to thymol and carvacrol, oregano contains a wide range of active compounds, including flavonoids and triterpenes exhibiting pronounced antioxidant activity [2–4]. Oregano is well-known in medicine due to its sedative, anxiolytic, and antispasmodic effects [5]. The Register of

Breeding Achievements Approved for Use in Russia includes 18 cultivars of oregano, 10 of which were registered in the period from 2000 to 2010 [6]. The eight patents belong to five state institutions; the other 10 belong to private companies located in different regions of the country.

Crimea is an important promising region for the cultivation of essential-oil-bearing and medicinal crops including oregano [7,8]. The optimal conditions for essential oil synthesis in various essential-oil-bearing crops are observed in this area.

The choice of a cultivar adapted to the agroclimatic conditions of the region is one of the main approaches for stable harvests' generation with specified quality characteristics. In Russia, there is no regionalization of new cultivars for the essential oil-bearing plants group. Therefore, it is important to consider the performance of cultivars from different breeding centers under specific soil and climatic conditions.

Various coefficients have been proposed and are widely used in agriculture to assess ecological adaptability, such as ecological variation, genotypic effect, plasticity, stability, homeostability, and breeding value [9–11]. Initially, they were used to evaluate the productivity of cereals. However, in recent years, researchers more often extrapolate them to different quantitative plant characteristics, not only yield [12,13].

However, there is no available literature on their use for essential-oil-bearing crops. Therefore, the authors consider that such study is valuable and relevant not only for Crimea, but for other regions as well.

In this regard, the objective of the research was to assess the possibility of using various indices of ecological adaptability originally developed for grain crops for their use in the most adapted genotypes' selection (breeding samples and cultivars) of *Origanum vulgare* L. to the temperate climate of the Crimean Peninsula.

## 2. Materials and Methods

The research was carried out from 2016 to 2019 in the piedmont zone of Crimea (Krymskaya Roza village, Belogorsky District, Russia, 45.053442 N, 34.361665 E). The soil at the experimental site was southern carbonate, heavy, loamy black soil with the following agrochemical characteristics: pH was from 7.0 to 7.2 (the essence of the method is to extract water-soluble salts from the soil with distilled water at a soil-to-water ratio of 1:5 and determine the pH level using a pH meter); humus content in the plow-layer was from 2.7 to 3.0% (the method is based on the organic matter oxidation with potassium dichromate solution in sulfuric acid and subsequent determination of trivalent chromium equivalent to the content of organic matter on a photoelectric colorimeter); nitrogen content was 0.12% (the method is based on soil sample mineralization when heated with concentrated sulfuric acid in the presence of hydrogen peroxide, followed by ammonia distillation into a boric acid solution and titration with sulfuric acid), phosphorus content was 0.1% (the method is based on soil sample mineralization when heated with concentrated sulfuric acid in the presence of hydrogen peroxide, followed by optical density determination of the colored phosphorus-molybdenum complex reduced to molybdenum blue); potassium content was 1.0% (the method is based on soil sample mineralization when heated with concentrated sulfuric acid in the presence of hydrogen peroxide, followed by optical density determination of the colored phosphorus-molybdenum complex reduced to molybdenum blue).

The meteorological conditions of the research years were assessed by the average daily air temperature (°C), the amount of precipitation per month (mm) according to the data of the meteorological station (Davis 6152EU Wireless Vantag Pro 2, stationary located N45o03′42″, E34o21′40″). The nursery was planted in 2016 and a comprehensive study was conducted during 2017–2019.

In 2017, significant deviations from long-term annual average temperatures were observed in March and August (mean air temperatures exceeded the long-term average by 4.0 and 2.1 °C, respectively). The spring–summer period was characterized by abundant precipitation. Only in March, precipitation was 11% less than the long-term average annual norm, while it was extremely high in April, May, and July, amounting to 215, 212, and

117% of the norm, respectively. The growing season in 2018 was characterized by increased average monthly temperatures, which differed from the long-term average values by 0.5–3.8 °C. The amount of precipitation since January was significantly lower than average. In April, when intensive growth and development of oregano plants starts, there was only 4.2 mm of precipitation, which makes up 12.4% of the long-term average. Precipitation in May and June was 98.8% and 39.3% of the norm. Excessive rainfall was observed only in July (169.8% of average precipitation). The above-mentioned weather conditions led to a faster course of the vegetative phases and, consequently, lower productivity of *O. vulgare*. The growing season in 2019 was characterized by above-average temperatures in March, May, and June, which differed from the long-term average (the norm) by 2.2–4.1 °C. June was particularly hot (+4.1 °C above normal). Starting from January, precipitation was significantly lower than the long-term average (the norm). Rainfall occurred in the latter half of June (averaging to about normal for the month) and in July (114.6% of the norm). The existing weather conditions were not favorable for normal plant development and the high yield; heavy rainfall in June and July prevented the normal process of flowering and the essential oil accumulation in the plants during this period. Due to these reasons, most essential oil crops had relatively low essential oil content in the raw material (Table 1).

Five oregano genotypes were used as the experimental material: breeding samples No. 10 and No. 82 from the collection of FSBSI "Research Institute of Agriculture of Crimea" and cultivars Zima, Raduga and Slavnitsa selected by the All-Russian Scientific Research Institute of Medicinal and Aromatic Plants. The plot area of each genotype was 3 m$^2$ with two replications, nutritional area of one plant—0.18 m$^2$ (0.3 × 0.6 m). Distribution was by randomized blocks.

**Table 1.** Meteorological conditions of the study period.

| Month | Temperature, °C | | | | | Precipitation | | | |
|---|---|---|---|---|---|---|---|---|---|
| | Minimum | Maximum | Average | Normal | Aberrance | Days | mm | Norm, mm | % of Norm |
| 2017 | | | | | | | | | |
| January | −13.6 | 12.6 | −1.7 | −0.8 | −0.9 | 19 | 46.0 | 41.2 | 110.0 |
| February | −11.8 | 15.6 | 1.2 | 0.0 | +1.2 | 8 | 37.0 | 32.9 | 112.5 |
| March | −1.0 | 21.6 | 7.9 | 3.2 | +4.0 | 13 | 27.0 | 41.6 | 64.9 |
| April | −2.5 | 25.7 | 9.4 | 10.0 | −0.6 | 10 | 73.0 | 39.2 | 186.2 |
| May | 3.0 | 25.6 | 15.2 | 14.9 | +0.3 | 10 | 72.0 | 47.4 | 151.9 |
| June | 7.7 | 34.0 | 20.1 | 18.5 | +1.6 | 8 | 57.0 | 68.5 | 83.2 |
| July | 11.9 | 36.9 | 22.9 | 22.3 | +0.6 | 7 | 55.0 | 53.4 | 103.0 |
| August | 10.3 | 37.9 | 24.1 | 20.2 | +2.1 | 9 | 19.0 | 47.9 | 39.7 |
| September | 6.3 | 23.8 | 16.8 | 15.2 | +1.6 | 5 | 1.0 | 36.9 | 2.7 |
| October | 2.7 | 30.5 | 11.8 | 10.2 | +1.6 | 14 | 54.0 | 37.1 | 145.5 |
| November | −6.0 | 20.5 | 7.1 | 5.3 | +1.8 | 12 | 44.0 | 48.6 | 90.5 |
| December | −2.9 | 20.6 | 7.7 | 1.6 | +6.1 | 16 | 39.0 | 55.0 | 70.9 |
| 2018 | | | | | | | | | |
| January | −16.3 | 14.8 | 0.8 | −0.8 | +1.6 | 10 | 24.0 | 41.2 | 58.3 |
| February | −10.6 | 13.2 | 1.7 | 0.0 | +1.7 | 8 | 18.6 | 32.9 | 56.5 |
| March | −7.3 | 19.4 | 5.5 | 3.2 | +2.3 | 9 | 17.6 | 41.6 | 42.3 |
| April | 0.4 | 26.6 | 13.8 | 10.0 | +3.8 | 2 | 4.2 | 39.2 | 10.7 |
| May | 4.9 | 28.8 | 18.0 | 14.9 | +3.1 | 6 | 33.6 | 47.4 | 70.9 |
| June | 5.2 | 34.2 | 21.3 | 18.5 | +2.8 | 4 | 22.8 | 68.5 | 33.3 |
| July | 14.7 | 34.3 | 23.1 | 22.3 | +0.8 | 13 | 79.8 | 53.4 | 149.4 |
| August | 11.7 | 33.6 | 23.6 | 20.2 | +3.4 | 1 | 3.0 | 47.9 | 6.3 |
| September | 1.9 | 33.7 | 17.9 | 15.2 | +2.7 | 7 | 43.0 | 36.9 | 116.5 |
| October | 2.8 | 23.8 | 13.3 | 10.2 | +3.1 | 8 | 24.4 | 37.1 | 65.8 |
| November | −5.8 | 17.2 | 4.1 | 5.3 | −1.2 | 7 | 31.0 | 48.6 | 63.8 |
| December | −4.7 | 9.4 | 1.3 | 1.6 | −0.3 | 20 | 67.2 | 55.0 | 122.2 |

**Table 1.** *Cont.*

| Month | Temperature, °C | | | | | Precipitation | | | |
|---|---|---|---|---|---|---|---|---|---|
| | | | | 2019 | | | | | |
| January | −9.3 | 18.4 | 2.3 | −0.8 | +1.5 | 11 | 20.4 | 41.2 | 49.5 |
| February | −11.2 | 18.1 | 1.8 | 0.0 | +1.8 | 8 | 18.2 | 32.9 | 55.3 |
| March | 5.2 | 20.3 | 5.4 | 3.2 | +2.2 | 6 | 7.8 | 41.6 | 18.8 |
| April | 3.9 | 24.7 | 9.5 | 10.0 | −0.5 | 9 | 17.4 | 39.2 | 44.4 |
| May | 4.8 | 31.6 | 17.3 | 14.9 | +2.4 | 5 | 20.0 | 47.4 | 42.2 |
| June | 12.3 | 32.6 | 22.6 | 18.5 | +4.1 | 8 | 66.2 | 68.5 | 96.6 |
| July | 11.6 | 32.6 | 21.8 | 22.3 | −0.5 | 7 | 61.2 | 53.4 | 114.6 |
| August | 12.0 | 33.7 | 22.3 | 20.2 | +2.1 | 5 | 11.0 | 47.9 | 23.0 |
| September | 5.1 | 31.7 | 17.3 | 15.2 | +2.1 | 4 | 48.4 | 36.9 | 131.2 |
| October | −1.2 | 27.8 | 13.0 | 10.2 | +2.8 | 6 | 17.8 | 37.1 | 48.0 |
| November | −5.8 | 25.4 | 8.9 | 5.3 | +3.6 | 4 | 16.4 | 48.6 | 33.7 |
| December | −7.3 | 17.4 | 5.0 | 1.6 | +3.4 | 8 | 33.6 | 55.0 | 61.1 |

Three traits of oregano were considered: the yield of fresh material, the mass fraction of essential oil per oven-dry mass, and the essential oil areal yield. During the experiment, counts and observations were carried out on 30 plants of each genotype in the phase of mass flowering starting from the second year of the growing season. For harvesting, all plants were cut from the plot at a height of 5–7 cm above the soil level. Then they were weighed with scales, the yield was recalculated for an area of 1 m$^2$. To determine the content of essential oil in the cut green raw material, a sample weighing 300 g was taken in two replicates in each repetition. The determination of the mass fraction of essential oil was carried out in freshly cut raw materials by the method of hydrodistillation according to Ginsberg [14]. To determine the mass fraction of essential oil in oven-dry raw materials, recalculation was performed considering the moisture content of the raw materials. Essential oil areal yield is the potential yield of essential oil per unit area that a given genotype can provide when grown. This value was calculated by multiplying the mass fraction of essential oil from the raw mass of raw materials by the potential yield of green mass per unit area. Six coefficients were used to calculate ecological adaptability, including a coefficient of ecological variation, genotypic effect, plasticity, stability, homeostability, and breeding value. These coefficients were calculated separately for each of the three traits (fresh yield, mass fraction of essential oil and its yield) for each genotype. All these coefficients were originally proposed to evaluate the productivity of grain crops [15–18]; but in this work, the authors attempted to apply them to some economically valuable traits of oregano.

The coefficient of ecological variation ($Cv_{ecol}$) is the variation of a characteristic by the year:

$$Cv_{ecol} = \frac{\sigma}{\overline{x}} \tag{1}$$

Homeostability (*Hom*) and breeding value (*Sc*) were determined according to Hangildin [15]:

$$Hom = \frac{\overline{x}^2}{\sigma} \tag{2}$$

$$Sc = \overline{x} \frac{\overline{x_{lim}}}{\overline{x_{opt}}} \tag{3}$$

where $\sigma$ is the standard deviation from the arithmetic mean of each genotype, $\overline{x}$ is the arithmetic mean for each genotype (for three research years) for the years of research, $\overline{x_{lim}}$ is the trait value for each genotype under limiting (worst) environmental conditions, $\overline{x_{opt}}$ is the trait value under optimal (best) environmental conditions.

Genotypic effect was calculated according to Guryev (methodological recommendations for ecological cultivar testing of corn, 1981) [16]:

$$E_i = \frac{\overline{x}_i}{\overline{x}_{cp}}$$

(4)

$\overline{x}_i$ is the average annual trait value of a particular cultivar, $\overline{x}_{cp}$. is the average trait value in all environments and samples.

Environmental plasticity and stability were calculated by the method of Eberhart and Russel [17]. The environment conditions were evaluated using the index $I_j$ and the linear regression coefficient ($b_i$) and the mean square deviation from the theoretical regression line ($\sigma d^2$) were calculated.

The linear regression coefficient of the studied trait of cultivars ($b_i$) was calculated by the formula:

$$b_i = \frac{\sum Y_{ij} \times I_j}{\sum I_j^2}$$

(5)

where $\sum Y_{ij} \times I_j$—the sum of the *i*-th grade characteristic value multiplication for the *j*-th year by the corresponding value of the index of environmental conditions; $\sum I_j^2$—sum of squares of environmental conditions' indices.

The index of environmental conditions $I_j$ was calculated by the formula:

$$I_j = \frac{\sum Y_{ij}}{v} - \frac{\sum \sum \sum Y_{ij}}{v \times n}$$

(6)

where $\sum Y_{ij}$—the sum of the characteristic values of all cultivars for the i-th year; $\sum Y_{ij}$– the sum of the characteristic values of all cultivars for all years; *v*—the number of cultivars; *n*—number of years.

The standard deviation (stability) is calculated by the formula:

$$\sigma d^2 = \frac{\sum \sigma d^2_{ij}}{n - 2}$$

(7)

where $\sum \sigma d^2_{ij}$—the sum of the squares of the actual value deviations of the trait from the theoretical value; *n*—number of testing years

The results were processed by analysis of variance in the Statistica 10.0 program, the evaluation of the influence strength of the factors was performed ($\eta^2$) according to Plokhinsky N.A. [18].

## 3. Results

The "year" factor had no significant influence on the yield of fresh material, while the influence of "genotype" and the interaction of "genotype × year" factors were found to be statistically significant. The mass fraction of essential oil and its areal yield were significantly influenced by genotype and year, and the effect of the interaction of "genotype × year" factors was within experimental error (Table 2).

The effect ($\eta^2$) of the studied factors (genotype, year, and their interaction) was calculated. The yield of fresh material was most strongly influenced by the genotype of the plant (43%), while the influence of the year conditions was at the level of 2%. On the contrary, the meteorological conditions had a much greater effect on the essential oil accumulation and its areal yield, amounting to 30% and 25%, respectively (Figure 1). Therefore, these two traits are more susceptible to environmental influences. This once again confirms the need for research on the particular genotype's adaptability when growing essential-oil-bearing crops not only in terms of yield, but also in terms of high essential oil content, as well as for assessing the genotype performance under different agroclimatic conditions.

**Table 2.** Results of a two-way analysis of variance of the main economic traits of oregano (2017–2019).

| Source of Variance | SS (Sum of Squares) | DF (Degrees of Freedom) | MS (Mean Square) | F |
|---|---|---|---|---|
| | | Yield of fresh material | | |
| Intercept term | 61.22 | 1 | 61.22 | 1547.95 * |
| Sample | 1.32 | 4 | 0.33 | 8.33 * |
| Year | 0.05 | 2 | 0.02 | 0.58 |
| Sample × year | 1.12 | 8 | 0.14 | 3.55 * |
| Error | 0.59 | 15 | 0.04 | |
| | | Mass fraction of essential oil | | |
| Intercept term | 0.960 | 1 | 0.960 | 92.152 * |
| Sample | 0.280 | 4 | 0.070 | 6.726 * |
| Year | 0.231 | 2 | 0.116 | 11.086 * |
| Sample × year | 0.105 | 8 | 0.013 | 1.259 |
| Error | 0.156 | 15 | 0.010 | |
| | | Essential oil areal yield | | |
| Intercept term | 0.233 | 1 | 0.233 | 77.349 * |
| Sample | 0.078 | 4 | 0.019 | 6.434 * |
| Year | 0.049 | 2 | 0.025 | 8.176 * |
| Sample × year | 0.025 | 8 | 0.003 | 1.043 |
| Error | 0.045 | 15 | 0.003 | |

* significant at $p \leq 0.05$.

Unlike essential oil crops and medicinal plants, many indices and criteria have been developed and tested for cereal crops, which can be used to assess the ecological adaptability of genotypes. In an attempt to express the ecological parameters of oregano genotypes in numerical terms, the following parameters were calculated: coefficient of ecological variation ($Cv_{ecol}$), genotypic effect ($E_i$), plasticity ($b_i$), stability ($\sigma d^2$), homeostability (*Hom*), and breeding value (*Sc*). The year of 2017 was the most favorable for the formation of high yield of fresh material, essential oil mass fraction and its areal yield, with environmental condition index $I_j$ = 0.11; 0.13 and 0.05, respectively. In the two other years, the environmental conditions indices for all three traits had negative values and ranged from $I_j$ = −0.02 to $I_j$ = −0.07.

The yield of fresh material in its physical meaning is the most similar to the original trait, for which the above coefficients were developed. Therefore, they can be considered separately without focusing on the actual values. The productive (1.48 kg/m²) collection sample No. 82 had the lowest coefficient of variation by year (Cv = 11.47%). According to the genotypic effect, the best were the Raduga cultivar ($E_i$ = 0.47) and the sample No. 82 ($E_i$ = 0.14). In terms of ecological plasticity, Raduga ($b_i$ = 4.52) and Zima ($b_i$ = 2.03) were assigned to the intensive type cultivars. Local collection sample No. 82 demonstrated the greatest stability in the fresh yield formation, $\sigma d^2$ was the lowest (0.008). The same genotype had the highest homeostability (*Hom* = 8.72) and breeding value (*Sc* = 1.62) (Table 3).

The essential oil yield and the mass fraction of essential oil traits significantly positively correlate (r = 0.97 on average over the research period) (Table 4).

Therefore, these two commercially valuable traits will be considered in close connection with each other.

The highest average actual value of the mass fraction of essential oil in oven-dry material was recorded for the local breeding sample No 82 (0.365) exceeding the other genotypes involved in research by 0.225–0.266%. This genotype was also the best according to breeding value, *Sc* = 0.22. This is confirmed by the fact that this sample exceeded the others in terms of essential oil yield (186 g/m²) by 0.116–0.140 g/m².

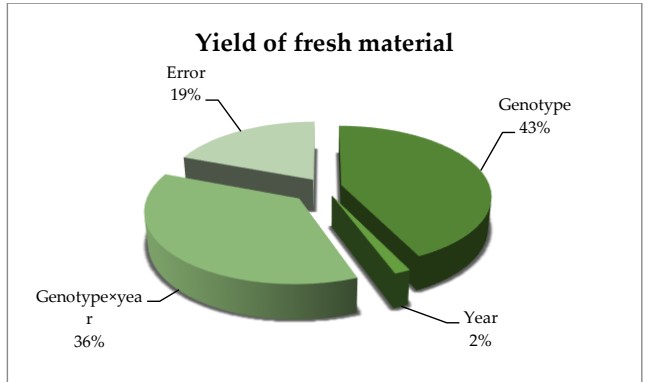

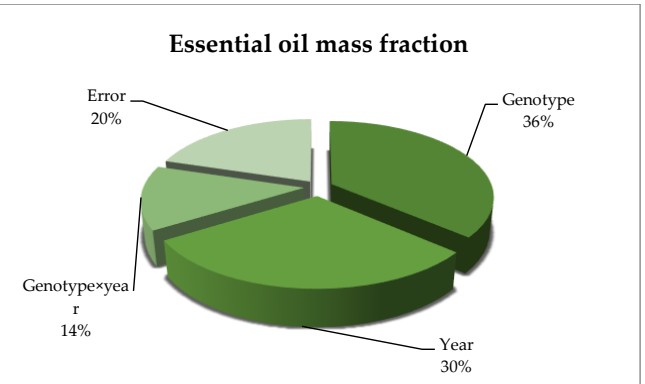

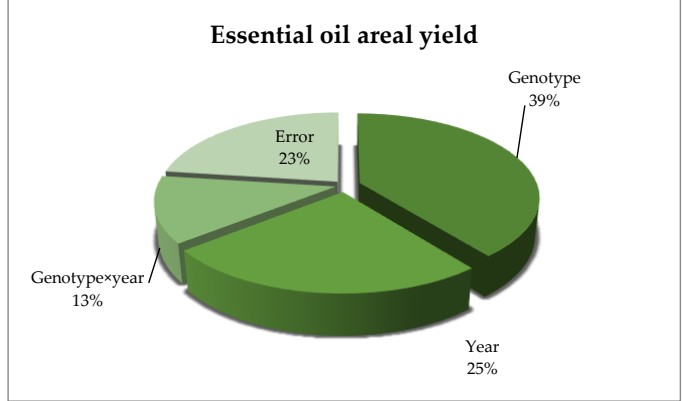

**Figure 1.** Effect size of the "genotype", "year" factors and their interaction on the yield of fresh material, the mass fraction of essential oil per oven-dry mass, and the essential oil areal yield.

**Table 3.** Parameters of ecological adaptability of oregano cultivars and breeding samples from different breeding zones in the conditions of the piedmont zone of Crimea.

| Cultivar | Coefficient of Ecological Variation ($Cv_{ecol}$), % | Genotypic Effect ($E_i$) | Plasticity ($b_i$) | Stability ($\sigma d^2$) | Homeostability ($Hom$) | Breeding Value ($Sc$) |
|---|---|---|---|---|---|---|
| | | | Yield of fresh material | | | |
| No 10 | 23.53 | −0.26 | 0.06 | 0.129 | 4.25 | 0.67 |
| No 82 | 11.47 | 0.14 | −1.73 | 0.008 | 8.72 | 1.62 |
| Slavnitsa | 16.70 | −0.20 | 0.10 | 0.072 | 5.99 | 1.59 |
| Zima | 21.24 | −0.15 | 2.03 | 0.058 | 4.71 | 1.60 |
| Raduga | 24.68 | 0.47 | 4.52 | 0.057 | 4.05 | 1.41 |
| | | | Essential oil content | | | |
| No 10 | 46.97 | −0.03 | 0.57 | 0.000 | 2.13 | 0.06 |
| No 82 | 52.24 | 0.20 | 1.47 | 0.017 | 1.91 | 0.22 |
| Slavnitsa | 124.57 | −0.03 | 1.50 | 0.002 | 0.80 | 0.00 |
| Zima | 107.81 | −0.07 | 0.89 | 0.002 | 0.93 | 0.00 |
| Raduga | 70.42 | −0.07 | 0.58 | 0.002 | 1.42 | 0.02 |
| | | | Essential oil yield | | | |
| No 10 | 51.53 | −0.03 | 0.46 | 0.000 | 1.94 | 0.02 |
| No 82 | 45.93 | 0.10 | 1.27 | 0.005 | 2.18 | 0.07 |
| Slavnitsa | 125.16 | −0.03 | 1.23 | 0.001 | 0.80 | 0.01 |
| Zima | 117.92 | −0.04 | 0.92 | 0.001 | 0.85 | 0.01 |
| Raduga | 91.52 | −0.01 | 1.12 | 0.001 | 1.09 | 0.02 |

**Table 4.** Correlation between essential oil yield and its components.

| Traits' Pair | 2017 | 2018 | 2019 | Average |
|---|---|---|---|---|
| Essential oil yield/Yield of fresh material | 0.07 | 0.43 * | 0.80 * | 0.20 |
| Essential oil yield/Essential oil mass fraction | 0.90* | 1.00 * | 0.96 * | 0.97 * |

* significant at $p \leq 0.05$.

The weather conditions of the year have a greater influence on essential oil content and its yield than on fresh yield, which is confirmed by the coefficient of ecological variation ($Cv_{ecol}$ was 45.97–24.57 and 45.93–125.16). According to the level of genotypic effect on both traits, the same collection sample No 82 stood out with Ei of 0.20 and 0.10, respectively. The Slavnitsa cultivar and sample No 82 can be classified as high-intensive in terms of essential oil content and essential oil yield ($b_i$ was 1.50 and 1.23 for Slavnitsa, 1.47 and 1.23 for No 82, respectively). The local genotypes were better than the introduced cultivars in terms of essential oil content and essential oil yield homeostability: sample No 10 had *Hom* = 2.13 and 1.94; sample No 82 had *Hom* = 1.91 and 2.18.

## 4. Discussion

The study involved valuable collection genotypes and cultivars of oregano of different ecological and geographical origin. This approach is applied for breeding of many crops [19,20]. To comprehensively assess the ability of a particular genotype to form economically important production in the research region, several parameters are calculated that reflect ecological adaptability [21–24] such as the coefficient of ecological variation, genotypic effect, plasticity, stability, homeostability, breeding value. As far as the authors are concerned, there are no publications on the possibility of using these coefficients for evaluating herbaceous raw material crops. It was shown that similarly to grain crops [17], genotypes with linear regression coefficient ($b_i$) greater than 1 and variance of deviations from the regression line ($\sigma d^2$) tending to 0 were classified as valuable genotypes by yield of fresh material, mass fraction of essential oil and essential oil yield [25]. In terms of the most valuable characteristic for essential oil-bearing oregano, the essential oil areal yield, the collection local genotype No. 82 and the cultivar Slavnitsa ($b_i$ = 1.23 . . . 1.50) were classified as such.

Homeostability (Hom) is a valuable property of a plant's self-regulation with the environment. The homeostasis effect allows to stabilize and maintain the normal reactions of the organism and to resist the limiting factors of the external environment [15,26]. This parameter is directly proportional to the sample yield and inversely proportional to its variation in different conditions [27]. Therefore, breeding samples and cultivars with high homeostaticity are of value. Collection sample No. 82 is valuable for cultivation in the conditions of the Crimean Peninsula according to the essential oil yield and two of its components—the yield of fresh material and the essential oil concentration (*Hom* was 8.72; 1.91; 2.18, respectively). Based on the long-term data analysis, the coefficient of ecological variation ($Cv_{ecol}$) makes it possible to assess the ecological stability and ecological influencing factors, as well as to determine the adaptivity of genotypes [28]. Its effectiveness has been shown for grain crops [25,29]. However, to assess the studied economically valuable characteristics of oregano, this coefficient was not sufficiently indicative with respect to other calculated parameters reflecting the ecological adaptability to the conditions of the region.

The genotypic effect ($E_i$) is determined by genotypic adaptation; it is a peculiar set of plant development programs that are inherited. The implementation of these programs depends largely on external conditions, which activate certain blocks of genes. The level of the studied traits' appearance of oregano under conditions differing in the ratio of moisture and temperature allowed to evaluate the genotypic effect of each sample. A high level of this trait's expression is characterized by a high positive value. This index is used to evaluate the genotypes of bean, pea, and corn [16,27,30]. In our study, all genotypes except No. 82 ($E_i$ = 0.10 . . . 0.20) and Raduga cultivar ($E_i$ = 0.47 by yield) were characterized by negative values for all three studied features ($E_i$ = −0.01 . . . −0.26), which defines them as having low adaptive capacity. This coefficient is ambiguous for use in evaluating herbaceous plants.

Breeding value (*Sc*) indicates the level of phenotypic stability of a particular sample and should approach a value of one. In this case, according to Hangildin [15], the genotype is not dependent on environmental conditions. However, for example, the analyzed

fresh herbal yield depends on several components and is controlled by the environmental conditions in which it is formed. According to the yield of fresh material when growing in the conditions of the Crimean Peninsula, the collection sample No. 82 of local origin was also the best with a breeding value of $Sc$ = 1.62 outperforming the rest of the genotypes included in the experiment.

Thus, the authors have shown the possibility of using coefficients characterizing adaptive cultivars' properties developed for grain crops' evaluation for oregano breeding samples and cultivars. According to the research results, it can be concluded that local genotypes are more adapted to the conditions of the region in terms of essential oil accumulation, but they are inferior to the cultivars included in the register in terms of fresh yield [8].

Since different directions of *O. vulgare* breeding are followed in Russian institutions, the assessment of the ecological adaptability of the crop would make it possible to recommend those or other cultivars depending on the purposes of cultivation [7].

## 5. Conclusions

The formation of oregano yield under the conditions of the Crimean continental climate was most strongly influenced by genotype (43%) with the influence of the weather conditions of the year at the level of 2%. On the contrary, the accumulation of essential oil and its areal yield were much more influenced by the weather conditions by 30% and 25%, respectively. According to the coefficient of ecological variation ($Cv_{ecol}$) in terms of fresh yield, sample No. 82 (11.47%) and cultivar Slavnitsa (16.7%) were the best. The local genotypes No. 10 and No. 82 were less variable in essential oil content and essential oil yield, exceeding the genotypes of the Moscow breeding center by 18–79% by this coefficient.

The genotypic effect size was greater than zero for yield in the Raduga cultivar and the local genotype No. 82, and only in the collection sample No. 82 for the other two traits. Zima and Raduga cultivars were classified as intensive according to ecological plasticity of yield with $b_i$ = 2.03 and 4.52, respectively, while the local genotype No. 82 and the Slavnitsa cultivar formed the intensive group according to essential oil content and essential oil yield with $b_i$ = 1.23–1.50.

The local genotypes were better than the introduced cultivars in terms of homeostability of essential oil content and essential oil yield, sample No. 10 had *Hom* = 2.13 and 1.94; sample No. 82 had *Hom* = 1.91 and 2.18 for these two traits, respectively.

**Author Contributions:** Conceptualization, E.M., A.M., N.N. and V.P.; methodology, S.B., L.R. and O.L.; software, S.B.; validation, E.M., S.B., A.M., N.N. and V.P.; research, E.M., A.M.; resources, O.L.; data curation, N.N.; writing—original draft preparation, E.M., S.B.; writing—review and editing, L.R., V.P., L.R.; visualization, S.B.; supervision, E.M.; project administration, V.P.; funding acquisition, V.P., O.L. All authors have read and agreed to the published version of the manuscript.

**Funding:** This research was carried out with the support of the Ministry of Education and Science of the Russian Federation under scientific projects No 0834-2019-0007/ AAAA-A16-116022610110-9.

**Data Availability Statement:** In this study, two *Origanum vulgare* L. breeding samples bred by the Research Institute of Agriculture of Crimea and three cultivars bred by All-Russian Scientific Research Institute of Medicinal and Aromatic Plants were used. The breeding samples were taken from the Collections of essential oil, spicy aromatic and medicinal plants (№507515 (http://www.ckp-rf.ru (accessed on 19 December 2021))); the cultivars were kindly provided by I. Korotkikh (All-Russian Scientific Research Institute of Medicinal and Aromatic Plants, Moscow, Russia).

**Conflicts of Interest:** The authors declare no conflict of interest.

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
