# Peer review of "Ecological Adaptability of Some Cultivars and Breeding Samples of Origanum vulgare L."

_agronomy, doi:10.3390/agronomy12010016_

Round 1
Reviewer 1 Report
The present study covers the research on the adaptability traits of selected cultivars and genotypes of Origanum vulgare L. According to my opinion, the newly obtained knowledge can be useful for the breeding programs and selection of proper genotypes and cultivars for industrial cultivation, for these reasons the manuscript could be suitable for publication after revisions. To improve the quality of the manuscript, I have certain questions as outlined below and suggest a few corrections:
- Lines 75-89 should be omitted.
- Lines 121-125 It is not clear how many exactly different genotypes were used. Please clarify and add information about the plant material used for the study. Indicate the genotypes. Were the five samples of each genotype used? When were the traits calculated?
- When was the plant material collected for the essential oil extraction? Which plant part was used? How was the mass fraction of essential oil determined? Yield? This should be indicated in the materials and methods section. Was the chemical composition of essential oils determined?
- Line 139 “where Ñ…Ì… is the average for the years of research”. Please indicate exactly what data is averaged. Is it the same for the formulas 1, 2 and 3?
- Line 152 the sentence is not finished.
- Line 158-159 should be included in materials and methods sections and clearly indicated which trait was used to corresponding formulas.
- Line 188 how this productive value was determined?
- Lines 239-240 please clarify. What is “bi”?
Author Response
Dear Reviewer.
Thank you for your attention to our work.
We agree with most of the comments and made changes to the text of the article. Let me give you an explanation for two remarks:
- Lines 121-125 It is not clear how many exactly different genotypes were used. Please clarify and add information about the plant material used for the study. Indicate the genotypes. Were the five samples of each genotype used? When were the traits calculated?
- Dear reviewer, by samples we meant breeding sample that we involved to the initial study. In the text of the article, we have replaced "sample" with "breeding sample". By a variety, we defined a genetically aligned genotype, entered in the register of cultivars, which is already cultivated. To avoid confusion, 'Variaty' has been replaced with 'Cultivar'. The 5 genotypes were studied during the experiment - 2 breeding samples and 3 included in the Register and allowed for cultivation in Russia cultivars. We hope that we have clarified the terminology, thank you for paying attention to this issue and we will not mislead potential readers.
- Line 158-159 should be included in materials and methods sections and clearly indicated which trait was used to corresponding formulas.
- All indices were calculated for each genotype according to the three most important traits of essential oil crops - the yield of fresh materials, the mass fraction of essential oil, and the essential oil areal yield.
Respectfully yours, team of authors.

Reviewer 2 Report
The manuscript Ecological adaptability of some varieties and samples of Ori-ganum vulgare L. presents interesting data, with potential interests from the readers of the journal. However, the presentation of the data is questionable. Materials and methods sections is completely inadequate, raising serious flaws if the data can be used or reproduced. The Results sections is difficult to follow because the authors did not use a standardized consequent terminology of the evaluated parameters.
Here is a detailed list of most important issues to be improved:
L14: L. should be added after vulgare
L17: Samples and varieties? What is the difference? Especially what are the characteristic of a sample?
L22: Formation of fresh yield? The authors mean ‘yield of fresh material’?
L23: Local conditions? What do the authors mean?
L40: Replace leaf crop by ‘herb crop’
L43: Italics for plant name
L44: in poultry and livestock as what?
L44-45: I would suggest the authors make the differentiation not between phenols and others, but between volatile fraction compounds (those extractible in the essential oils) and so-called fixed components (flavonoids, polyphenols, phenolic acids, triterpenes).
L48-49: I am not following the relevance of this statement.
L56: The authors need to make the differentiation between ‘essential oil’, which is a type of extract (usually obtained via hydro or steam distillation) and ‘volatile fraction’ which represent the compounds that are extractible in the essential oil. Therefore, ‘accumulation of essential oil’ in plant is not a correct terminology.
L70-72: The aim of the study need to be extended, to include all the research aspects the authors have addressed in this paper.
L74: The materials and methods sections needs to be totally restructured and presented in a scientific manner. Subtitles are needed for each parameter determined/assay together with the accurate, reproducible description of the Methods.
No information on how the authors extracted the volatile compounds and how they analyzed the essential oils.
L75-89: Needs to be deleted.
L92-94: This is a Results section. In materials and methods, we need to know how all these parameters were obtained? How did you determine the ph, the humus content, the nitrogen content, etc?
L98-120: Same as above. It is more a description of the conditions. It is not clear from where the authors took this data (Table 1)? From weather forecasts? OR were they monitored by the authors? If yes, how, which methodologies have been used?
L121-122: You say five oregano samples… samples no. 10 and no. 82 and three varieties. You have to differentiate by samples and varieties. You have two samples and three varieties, so practically five samples, indeed. But it is confusing. The authors need to make a clear differentiation.
L129: References for grain crops is missing.
L131: Please use the superscript as in equation one for Cvecol
L133: Define S and x in Eq. 1
L158-159: How were these parameters calculated? You need to provide equations in the material and methods, especially since you have three types of yields (Areal yield, fresh yield, oil yield)? Differences?
), it is difficult to follow which is which. A standardized clear consequent terminology over the manuscript is missing.
L159: Mass fraction of essential oil per absolutely dry mass is practically the ‘essential oil yield’ (while in the abstract you used just ‘oil yield’).
L160: Which yield you refer here?
L162: Areal yield? How did you define it? It is also in relation to essential oil, right?
L168: Which yield of those many types you have?
L170: Essential oil accumulation? Is this also essential oil yield? As mentioned in one of my above content, the essential oil is not accumulating, since does not exist in the plant. I see from table 4, that actually you have essential oil yield, fresh yield, essential oil mass fraction. No mention about the areal yield anymore.
L221: Suddenly you use EOC abbreviation, that you did not use it since abstract.
Author Response
Thank you for your attention to our work.
We agree with most of the comments and made changes to the text of the article. Let me give you an explanation for two remarks:
L17: Samples and varieties? What is the difference? Especially what are the characteristic of a sample?
- Dear reviewer, by samples we meant breeding sample that we involved to the initial study. In the text of the article, we have replaced "sample" with "breeding sample". By a variety, we defined a genetically aligned genotype, entered in the register of cultivars, which is already cultivated. To avoid confusion, 'Variaty' has been replaced with 'Cultivar'. The 5 genotypes were studied during the experiment - 2 breeding samples and 3 included in the Register and allowed for cultivation in Russia cultivars. We hope that we have clarified the terminology, thank you for paying attention to this issue and we will not mislead potential readers.
L23: Local conditions? What do the authors mean?
- By local conditions, we understood the soil and climatic conditions in the study region. In our case, this is the Crimean Peninsula.
- 48-49. L56: The authors need to make the differentiation between ‘essential oil’, which is a type of extract (usually obtained via hydro or steam distillation) and ‘volatile fraction’ which represent the compounds that are extractible in the essential oil. Therefore, ‘accumulation of essential oil’ in plant is not a correct terminology.
- Since essential oils are secondary metabolites of the vital activity of the plant organism, which are deposited (accumulate) in specialized containers (glands and glandular hairs). We replaced the term accumulation with synthesis. We hope this will be more accurate. In our study and in well-researched literature sources, the content of essential oil is understood as the amount of essential oil that is isolated by the method of hydrodistillation.
L92-94: This is a Results section. In materials and methods, we need to know how all these parameters were obtained? How did you determine the ph, the humus content, the nitrogen content, etc?
- We expanded by specifying the techniques. However, we believe that this information relates to the conditions of the experiments, so that readers can extrapolate our results to their region. This information does not help in any way to reveal the purpose, but only acquaints the reader with the characteristics of the soil where the studied genotypes of oregano were grown.
L98-120: Same as above. It is more a description of the conditions. It is not clear from where the authors took this data (Table 1)? From weather forecasts? OR were they monitored by the authors? If yes, how, which methodologies have been used?
- We clarified the brand of the meteorological station, with the help of which the ambient temperature and precipitation and its coordinates were estimated. However, we believe that this information relates to the conditions of the experiments, so that readers can extrapolate our results to their region. This information does not help in any way to reveal the purpose, but only acquaints the reader with the characteristics of the meteorological conditions where the studied genotypes of oregano were grown.
L121-122: You say five oregano samples… samples no. 10 and no. 82 and three varieties. You have to differentiate by samples and varieties. You have two samples and three varieties, so practically five samples, indeed. But it is confusing. The authors need to make a clear differentiation.
- Dear reviewer, by samples we meant breeding sample that we involved to the initial study. In the text of the article, we have replaced "sample" with "breeding sample". By a variety, we defined a genetically aligned genotype, entered in the register of cultivars, which is already cultivated. To avoid confusion, 'Variaty' has been replaced with 'Cultivar'. The 5 genotypes were studied during the experiment - 2 breeding samples and 3 included in the Register and allowed for cultivation in Russia cultivars. We hope that we have clarified the terminology, thank you for paying attention to this issue and we will not mislead potential readers.
L160: Which yield you refer here?
- yield of fresh material.
L168: Which yield of those many types you have?
- yield of fresh material.
L170: Essential oil accumulation? Is this also essential oil yield? As mentioned in one of my above content, the essential oil is not accumulating, since does not exist in the plant. I see from table 4, that actually you have essential oil yield, fresh yield, essential oil mass fraction. No mention about the areal yield anymore.
- Essential oil areal yield is the potential yield of essential oil per unit area that a given genotype can provide when grown. This value was calculated by multiplying the mass fraction of essential oil from the raw mass of raw materials by the potential yield of green mass per unit area.
Respectfully yours, team of authors.

Round 2
Reviewer 1 Report
Line 98: why did you delete geographical coordinates?
Line 160: Method of Ginsburg. Add citation.
Line 210: ∑ σд or ∑ σd? unify the symbols used in formulas.
Author Response
Dear Reviewer.
Thank you for your attention to our work.
We agree with all of the comments and made changes to the text of the article.
Regards, team of authors.

Reviewer 2 Report
The authors considered most of the suggested recommendations in the revised versions
Author Response
Thank you for your attention to our work.
We've made changes to English.
Regards, team of authors.
